# Low Frequency of Cancer-Predisposition Gene Mutations in Liver Transplant Candidates with Hepatocellular Carcinoma

**DOI:** 10.3390/cancers15010201

**Published:** 2022-12-29

**Authors:** Klara Horackova, Sona Frankova, Petra Zemankova, Petr Nehasil, Marta Cerna, Magdalena Neroldova, Barbora Otahalova, Jan Kral, Milena Hovhannisyan, Viktor Stranecky, Tomas Zima, Marketa Safarikova, Marta Kalousova, CZECANCA Consortium, Jan Novotny, Jan Sperl, Marianna Borecka, Sandra Jelinkova, Michal Vocka, Marketa Janatova, Petra Kleiblova, Zdenek Kleibl, Milan Jirsa, Jana Soukupova

**Affiliations:** 1Institute of Medical Biochemistry and Laboratory Diagnostics, First Faculty of Medicine, Charles University and General University Hospital in Prague, 12808 Prague, Czech Republic; 2Department of Hepatogastroenterology, Institute for Clinical and Experimental Medicine, 14021 Prague, Czech Republic; 3Institute of Pathological Physiology, First Faculty of Medicine, Charles University and General University Hospital in Prague, 12853 Prague, Czech Republic; 4Department of Paediatrics and Inherited Metabolic Disorders, First Faculty of Medicine, Charles University and General University Hospital in Prague, 12808 Prague, Czech Republic; 5Centre for Experimental Medicine, Institute for Clinical and Experimental Medicine, 14021 Prague, Czech Republic; 6Department of Biochemistry, Faculty of Natural Science, Charles University, 12800 Prague, Czech Republic; 7Institute of Biology and Medical Genetics, First Faculty of Medicine, Charles University and General University Hospital in Prague, 12800 Prague, Czech Republic; 8Department of Internal Medicine, First Faculty of Medicine, Charles University and Military University Hospital, 16902 Prague, Czech Republic; 9Department of Oncology, First Faculty of Medicine, Charles University and General University Hospital in Prague, 12808 Prague, Czech Republic

**Keywords:** hepatocellular carcinoma, liver cirrhosis, liver transplantation, genetic predisposition, panel sequencing, MRN complex, germline mutation

## Abstract

**Simple Summary:**

Hepatocellular carcinoma (HCC) is the fourth most common cause of cancer-related deaths worldwide. HCC mostly results from liver cirrhosis and its genetic predisposition is believed to be rare. A liver transplantation is considered a curative therapy for HCC; however, de novo tumor development is a feared complication in immunosuppressed transplant recipients. Having analyzed the prevalence of pathogenic/likely pathogenic germline variants in cancer-predisposition genes in 334 HCC patients considered for liver transplantation, we found only 7/334 (2.1%) carriers of pathogenic variants in established cancer-predisposition genes (*PMS2*, 4×*NBN*, *FH* or *RET*). Interestingly, two MRN complex genes (*NBN* and *RAD50*) were significantly more frequent among patients over controls. Therefore, we conclude that the genetic predisposition to HCC is rare and HCC does not meet the criteria for routine germline genetic testing; however, germline testing could be considered in liver transplant recipients as the variant carriers may benefit from tailored follow-up or targeted therapy.

**Abstract:**

Hepatocellular carcinoma (HCC) mainly stems from liver cirrhosis and its genetic predisposition is believed to be rare. However, two recent studies describe pathogenic/likely pathogenic germline variants (PV) in cancer-predisposition genes (CPG). As the risk of de novo tumors might be increased in PV carriers, especially in immunosuppressed patients after a liver transplantation, we analyzed the prevalence of germline CPG variants in HCC patients considered for liver transplantation. Using the panel NGS targeting 226 CPGs, we analyzed germline DNA from 334 Czech HCC patients and 1662 population-matched controls. We identified 48 PVs in 35 genes in 47/334 patients (14.1%). However, only 7/334 (2.1%) patients carried a PV in an established CPG (*PMS2*, 4×*NBN*, *FH* or *RET*). Only the PV carriers in two MRN complex genes (*NBN* and *RAD50*) were significantly more frequent among patients over controls. We found no differences in clinicopathological characteristics between carriers and non-carriers. Our study indicated that the genetic component of HCC is rare. The HCC diagnosis itself does not meet criteria for routine germline CPG genetic testing. However, a low proportion of PV carriers may benefit from a tailored follow-up or targeted therapy and germline testing could be considered in liver transplant recipients.

## 1. Introduction

Hepatocellular carcinoma (HCC) is the fourth most frequent cause of cancer-related deaths and the fifth most frequent malignancy globally [1]. With 854,000 new cases and 810,000 deaths annually, HCC represents 7% of all malignancies. Diagnosis of HCC is responsible for 90% of primary liver tumors [2]. Its incidence increases with age and peaks at the age of 70; however, the age at diagnosis is significantly lower in Chinese and black African populations. Males are affected 2–2.5× more often than females. The incidence also varies geographically, with the highest incidence reported in low- and middle-resource countries from Southeastern Asia and Sub-Saharan Africa, accounting for more than 85% of the new global cases of HCC. In Europe, the incidence is significantly lower except for Southern Europe [3].

Approximately 90% of HCC cases occur in cirrhotic liver patients associated with chronic hepatitis B or C; alcoholic or metabolic liver disease, including non-alcoholic steatohepatitis (NASH); hereditary hemochromatosis or alpha-1-antitrypsin deficiency [2]. One-third of patients with liver cirrhosis develop HCC. The annual risk of HCC development in cirrhotic patients is estimated to be 1–8%, depending on the liver disease severity [4,5]. Liver transplantation represents the curative therapy with the best long-term results [6]. The 1-year and 5-year survival rates of liver transplant recipients are 90% and 70%, respectively, with de novo malignancies being the most frequent cause of late mortality in immunosuppressed liver transplant recipients [7]. To reduce mortality, guidelines for prevention and management of de novo tumors have been published recently [6].

In contrast to other cancer types, the hereditary component of HCC is considered rare [8]. However, recently published studies revealed that 11.4–12.6% of HCC patients carried pathogenic/likely pathogenic germline variants (PV) in some cancer-predisposition genes (CPG), including established high-penetrant genes causing hereditary breast/ovarian cancer (*BRCA1*, *BRCA2, PALB2*) or Lynch syndrome (*MLH1*, *MSH2, MSH6*) [9,10].

We hypothesized that the risk of de novo malignancies after liver transplantation might be increased in immunosuppressed PV carriers in CPGs. To this end, we aimed to identify the prevalence of PV in a retrospective, precisely clinicopathologically characterized single-center cohort of 334 consecutive liver transplant candidates with HCC in this study.

## 2. Materials and Methods

### 2.1. Patients

The study group consisted of 334 HCC patients (258 males and 76 females) referred to the Institute for Clinical and Experimental Medicine in Prague as liver transplantation candidates between August 2002 and September 2021. In the majority of patients (329/334, 98.5%), liver cirrhosis was diagnosed in accordance with the diagnostic guidelines before HCC onset [3]. There was no evidence of liver cirrhosis found for only five HCC patients. The etiology of the liver cirrhosis was based on patients’ medical history and laboratory data (Table 1).

Regarding the treatment modalities, 299 patients underwent liver transplantation, 34 patients were referred to palliative oncological therapy or best supportive care, and a single patient underwent liver resection. The median follow-up was 4.2 years (range 0.1–22.2 years). Demographic, laboratory and histopathological data were extracted from the hospital electronic information system. All but two patients were Caucasian of Czech origin, gave written informed consent to storing of their blood samples, and agreed to use of the blood samples for future research, including genetic testing. 

### 2.2. Controls

Data from two population-matched control groups were used for germline variant evaluation. For variant prioritization, we used a group of “super-controls” consisting of 791 healthy, non-cancer, older individuals aged >60 years (92 males and 697 females), without personal and first-degree family member cancer history. For case-control analyses, we used an independent control group consisting of 1662 unselected population-matched controls (PMC) provided by the National Center for Medical Genomics (http://ncmg.cz, accessed on 1 April 2022), in details described previously [11].

### 2.3. Library Preparation, Sequencing and Bioinformatics

Patients’ genomic DNA was isolated from peripheral blood using the Qiagen QIAamp DNA blood kit (Qiagen, Hilden, Germany). One hundred ng of gDNA was processed for the NGS library preparation using a KAPA HyperPlus Kit (Roche, Basel, Switzerland) according to the manufacturer’s instructions. Briefly, gDNA was enzymatically fragmented for 12.5 min at 37 °C, targeting 200 bp DNA fragments. The preparation of libraries, including the use of in-house-designed adapters and dual index primers used in a six-cycle, ligation-mediated polymerase chain reaction (LM-PCR), as well as the primers subsequently used in post-capture PCR, have been described in Soukupova et al. [12]. The prepared pre-library was eluted to a final volume of 30 µL, checked with a High Sensitivity DNA kit using a Bioanalyzer 2100 (both from Agilent Technologies, Santa Clara, CA, USA) and quantified with dsDNA High Sensitivity Assay Kits (Qubit assays) using a Qubit Flex Fluorimeter (both from Thermo Fisher Scientific, Waltham, MA, USA). Seventy-two barcoded samples were equimolarly pooled yielding a 1.5 µg DNA pool in 45 µL volume, concentrated if necessary. Pooled samples were then hybridized at 55 °C for 16–20 h using a KAPA HyperCapture Reagent kit from Roche and a Roche-made custom-designed CZECANCA (CZEch CAncer paNel for Clinical Application) panel capturing 226 established and candidate CPG [12] (Appendix A). The post-hybridization clean-up and amplification (in 11 cycles) were performed according to the manufacturer’s instructions for <40 Mbp capture target size with minor workflow modifications, including the in-house-designed post-capture PCR primers, as described previously [12]. The final library concentration was measured with the Qubit dsDNA HS Assay Kit and the targeted gene enrichment was checked using qPCR with in-house designed primers (available upon request). Finally, two libraries (each consisting of 72 samples) were proportionally pooled together at a final 1.5 pM concentration and prepared for NextSeq sequencing by adding 0.03 pM PhiX. Sequencing was performed on an Illumina NextSeq 500 instrument using the NextSeq 500/550 Mid Output Kit v2.5 for 150 cycles (Illumina, San Diego, CA, USA).

The sequencing data stored as FASTQ files were generated from NextSeq using an Illumina BaseSpace Sequence Hub and processed as described in Soukupova et al. [12] with minor upgrades. Briefly, the FASTQ files were mapped to a reference genome hg19 using Novoalign v2.08.03 (http://www.novocraft.com/products/novoalign/, accessed on 1 April 2022), providing the corresponding SAM and afterward BAM files using Picard tools v1.129 (https://broadinstitute.github.io/picard/, accessed on 1 April 2022). The BAM files served for identification of single nucleotide variants (SNV), medium-size insertions and deletions (indels), as well as copy number variants (CNV). For SNV analyses, VCF files were generated from the BAM files (following the exclusion of PCR duplicates) using GATK toolkit v3.8.1 (https://gatk.broadinstitute.org/, accessed on 1 April 2022) [13] and annotated with SnpEff v4.3 (https://pcingola.github.io/SnpEff/, accessed on 1 April 2022) [14]. Medium-sized indels were characterized using Pindel (http://gmt.genome.wustl.edu/packages/pindel/, accessed on 1 April 2022) [15] and CNV were identified with CNVkit (https://pypi.org/project/CNVkit/, accessed on 1 April 2022), as described previously [12].

### 2.4. Variant and Gene Prioritization

The variant prioritization aimed to identify clinically significant PVs. From the raw called variants, we sequentially filtered out variants (i) with low sequencing quality (<150); (ii) localized in repetitive and low-complexity DNA sequences (using RepeatMasker [16]); and (iii) non-coding (3’/5’UTR, downstream/upstream/intergenic/intragenic/deep intronic) variants and in-frame indels. Further, we filtered out variants with low clinical impact including those present (iv) in the group of super-controls with minor allele frequency (MAF) > 0.4%; (v) in the general population (gnomAD, 1000 Genomes Project, NHLBI GO ESP, ExAC databases [17,18,19,20]) with MAF > 0.4%; and (vi) interpreted as benign/likely benign (B/LB) by ClinVar [21]. Additionally, we excluded variants (vii) in last exons; (viii) in introns out of conserved splice site (>2 bp from an exon boundary); (ix) synonymous variants and (x) sequencing errors except for known PV. Finally, we filtered out variants without ClinVar interpretation as pathogenic/likely pathogenic, unless they caused premature termination, frameshift or aberrant splicing (1–2 bp from an exon).

Identified PV were confirmed using Sanger sequencing and/or MLPA and divided as (i) variants in established high-to-moderate CPGs (N = 48; including genes with germline variants of probable prognostic or predictive potential) or (ii) candidate CPGs (N = 178; including genes with uncertain prognostic or predictive effects of their germline variants; Figure 1 and Appendix A).

### 2.5. Statistical Analyses

Student’s t-test or the non-parametric Mann–Whitney and Kruskal–Wallis tests were used for continuous data, and categorical data were analyzed using the chi-square test. The survival rates were assessed with Kaplan–Meier analysis and the log-rank test was used to compare survival rates between individual groups. All statistical analyses were two-sided and a *p*-value of <0.05 was considered statistically significant. Statistical analysis was performed using the GraphPad Prism 9.3.1 software (GraphPad). Risk scores for PV carriers in HCC patients vs. PMC were calculated as odds ratio (OR) and 95% confidence interval (95% CI). 

## 3. Results

### 3.1. Germline Variants in Established and Candidate CPG

Altogether, 334 patients’ DNA samples were sequenced with a mean coverage of 119× enabling reliable copy number variant (CNV) calling. Identified variants were prioritized as described in the Methods section, yielding 48 PV in 35 genes found in 47/334 (14.1%) patients (Table 2). However, only 7/334 (2.1%) patients harbored a PV in 4/48 established high-to-moderate CPGs, including *PMS2, NBN, FH* and *RET* (Table 2). The most frequent was a frameshift variant c.657del5 (c.657_661delACAAA) in *NBN* found in four patients. The remaining PVs included a novel 8907 bp deletion affecting exons 11–12 in *PMS2* (Figure 2) and missense PVs in *FH* and *RET*. In 1662 PMCs, we detected four carriers of *PMS2* and *NBN* variants, respectively, two carriers of PVs in *RET* and none in *FH.* However, a statistically significant difference in frequency was observed only for *NBN* (*p* = 0.012; Table 3). 

In addition, we detected PVs in 31/178 candidate CPGs in 40/334 (12.0%) patients (including a patient harboring simultaneous germline variants in *ATRIP* and *RAD50*; Table 2). Overall 104/1662 (6.3%) individuals among PMC carried germline variants in these 31 genes (Table 3). Germline variants in HCC patients were recurrently found in only eight candidate genes including *DMBT1*, *RAD50, ATRIP*, *BLM*, *ERCC2*, *LIG3*, *MSH3* and *SLX4*; however, only *DMBT1*, *RAD50* and *LIG3* germline variants showed a significant difference in HCC patients compared to PMC (Table 3). 

Notably, PVs in seven HCC patients affected the genes coding for proteins of the MRN (MRE11-RAD50-NBN) complex, including four carriers of c.657del5 in *NBN* and three carriers of different variants in *RAD50* (Table 2). PVs in *NBN* and *RAD50* were significantly enriched in analyzed HCC patients over PMC (7/334; 2.1% vs. 7/1662; 0.4%; *p* = 0.001).

### 3.2. Clinical Characterization of PV Carriers

Patients with PVs in established CPGs, candidate CPGs, or in MRN complex genes differed from the variant non-carriers neither in demographic characteristics (age, cause of cirrhosis or occurrence of HCC in non-cirrhotic liver, diabetes or obesity), nor in tumor characteristics (angioinvasion, cholangiogenic differentiation, recurrence after liver transplantation). Moreover, the variant carriers did not present an increased frequency of multiple primary tumors (either before or after the liver transplantation) or a higher rate of primary malignancies in their first-degree relatives (Appendix A). The survival of patients was comparable between non-carriers and carriers of established CPG, candidate CPG and MRN genes (Appendix A).

## 4. Discussion

In this single-center study, we performed germline genetic testing on 334 patients with HCC indicated for liver transplants. PVs in the analyzed genes were found in 47/334 (14.1%) patients; however, only 7/334 patients (2.1%) carried a PV in established high-to-moderate CPGs. Of these genes, only variants in *FH* can be considered as high-penetrant and were previously described in HCC patients [10]. Moreover, *NBN* was the most frequently altered gene (Table 3) with four identified carriers of a recurrent Slavic c.657del5 variant [22] that moderately increases the risk of various cancer types in our population [11,23]. The *NBN* gene encodes for a protein stabilizing the MRN complex that regulates double-stranded DNA break repair [24]. Interestingly, we also identified three HCC patients who carried a PV in *RAD50* encoding another MRN complex protein. Thus, in total, seven (2.1%) HCC patients carried a PV in MRN complex genes compared to only 7/1662 (0.4%) controls (*p* = 0.001). While PV carriers in *NBN* and *RAD50* were observed also in previous HCC studies (Table 4), none was found in *MRE11*, the third gene of the MRN complex; however, its germline variants are rare [25]. Interestingly, germline variants in *NBN* were linked to HCC susceptibility in cirrhotic patients with chronic HBV infection previously [26,27]. In animal models, an increased formation of liver tumors was observed in mice hemizygous for the *Nbn* gene [28]. These findings suggest the possible involvement of the MRN complex in HCC development; however, further research, including mechanistic studies of HCC pathogenesis and large analyses in HCC patients are required.

The overall frequency of PV carriers in our HCC patients (14.1%) corresponds to the results published previously (Table 4) by Mezina et al. [10], who identified 25/217 (11.5%) carriers in prospective and 30/219 (13.7%) in retrospective cohorts of HCC patients. Another small study by Uson Junior et al. identified seven (15.9%) PVs in a set of 44 HCC patients [9]. However, the panel of genes analyzed in these studies varied, with ours being the largest (Table 4). The proportion of deleterious variants declined when only PVs in high-to-moderate CPGs were considered (Table 4). However, unlike ours, Mezina et al.’s retrospective study identified nine patients with germline *BRCA1*/*BRCA2* variants (entirely absent in our cohort) and four patients with germline alterations in Lynch syndrome genes.

The varying frequencies of PV carriers (2.1–11.4%) in high-to-moderate CPGs in the abovementioned studies reflect different enrollment criteria and diverse characteristics of the HCC cohorts. While HCC patients in three studies (this report, the prospective arm of Mezina et al.’s study, andUson Junior et al.’s study) were first enrolled and germline genetic testing was performed subsequently, individuals with the HCC diagnosis were selected retroactively from a large dataset of patients (analyzed in the commercial laboratory; Invitae) in the retrospective arm of the study by Mezina et al. Prospective studies were characterized by a low frequency of PVs in the genes conferring high overall cancer risk (*APC*, *BRCA1*, *BRCA2*, *PALB2,* Lynch syndrome genes) that are routinely tested for hereditary cancer syndromes (Table 4). In contrast, carriers of PVs in such genes were enriched in the retrospective (Invitae) cohort in Mezina et al.’s study [10]. We speculate that the HCC diagnosis among carriers from this retrospective cohort may represent a confounding event in individuals with HCC risk factors (alcohol abuse, HBV/HCV infections, etc.). Additionally, compared to our data (Appendix A), the HCC patients in Mezina et al.’s study are characterized by a high frequency of individuals with second primary tumors (17.1 vs. 38.4% in the prospective study) and a high frequency of analyzed patients with positive family cancer history (39.2 vs. over 80% in both prospective and retrospective studies). Also, the retrospective study of Mezina and colleagues included an unusual proportion of female patients compared to their prospective study and our report (56.2 vs. 16.8 and 22.6%, respectively). Moreover, it is possible that the proportion of PV carriers in highly penetrant genes in our study is artificially lower due to the potential early onset of their first cancer before HCC (median age of our cohort is 63 years). Thus, these PV carriers would develop HCC as their second tumor and, hence, they would not be eligible for liver transplantation, referred to the specialized tertiary care center and included in our study.

For additional evidence, we looked for HCC patients from the Czech CZECANCA consortium database [29]. Among 10,480 cancer patients, we identified 20 individuals with HCC diagnosis of which two were PV carriers in established CPGs (*BRCA1* and *CHEK2*; Appendix A). These findings resemble results from Mezina et al.’s retrospective study [10] indicating that PVs in HCC patients are likely found incidentally and can hardly be considered a genetic cause of HCC. It is of note that the risk for HCC development has not been estimated (or even documented) for any of the CPGs mentioned in this report. The results of our study support previous assumptions expecting a low hereditary component of HCC.

Mezina et al. also suggested germline variants in *FANCA* and *BRIP1* as candidates for HCC susceptibility. While the frequency of *FANCA* variants was comparable among our HCC patients and PMC, *BRIP1* variants were not detected in our study. Moreover, we found rare germline variants in *PMS1* [30], and other DNA damage response (DDR) genes *ERCC2* and *XRCC1* (associated with an increased risk of liver cirrhosis and its potential transformation into HCC in HBV-positive patients) [26,27], but we failed to identify PVs in other CPGs (including *BAP1*, *DICER1*, *HNF1A*, *MET*, *TERT* and *VHL*) associated with HCC in other studies [31,32,33,34,35,36].

Concerning the clinicopathological characteristics, only 5/334 individuals in our cohort developed HCC in the non-cirrhotic liver, corresponding to an expected causal effect of cirrhosis on HCC development. None of the non-cirrhotic patients carried a PV in the analyzed genes. Due to the low overall frequency of variant carriers, we did not notice any considerable differences in the carriers’ clinicopathological or tumor characteristics compared to the non-carriers.

Despite the low frequency of germline variants, germline genetic testing of HCC patients could be a prospect for precision medicine or targeted therapy. The PV carriers in Lynch syndrome genes and *BRCA1*/*BRCA2* could benefit from treatment with immune checkpoint (PD-1/PD-L1 inhibitors) and PARP (PARPi) inhibitors, respectively [9,10]. Moreover, a widening PARPi indication could include the PV carriers in the MRN complex and/or other DDR genes [37]. Genetic testing might be of particular importance in a subgroup of HCC patients indicated for liver transplantation. The high lifetime cancer risk in PV carriers in CPGs could strongly accelerate the development of de novo tumors in immunosuppressed transplant recipients. Several such cases have been reported in individuals with various organ transplantation episodically [38,39,40,41,42], but a systematic study in liver transplantation recipients is still missing. Our study indicates that a larger cohort of HCC patients indicated for liver transplantation will be required to perform such analysis due to the low frequency of PV carriers in CPGs among the patients. However, it must be stressed that genetic testing results must not influence liver transplantation eligibility.

The strength of this study includes the rigorous enrollment of well-characterized HCC patients unbiased from the recruitment of patients indicated for germline genetic testing. The study was limited by the predominance of younger liver transplant candidates with less advanced HCC, complying with the criteria for liver transplantation. The germline genetic testing was limited to the 226 cancer-predisposition genes included in the CZECANCA panel, which was designed for the analysis of cancer predisposition but does not cover some of the known cirrhosis-predisposing genes (i.e., *APOB*, *HFE*, *PNPLA3*, *SERPINA1*) [43,44,45,46,47].

## 5. Conclusions

We conclude that the low overall prevalence of PV carriers makes germline genetic testing in HCC diagnosis rather unnecessary unless the patients fulfil other criteria for germline genetic testing (including the presence of indicative second primary tumors or positive family cancer history). However, germline genetic testing might be considered for liver transplant recipients to reduce late mortality from de novo malignancies.

## Figures and Tables

**Figure 1 cancers-15-00201-f001:**
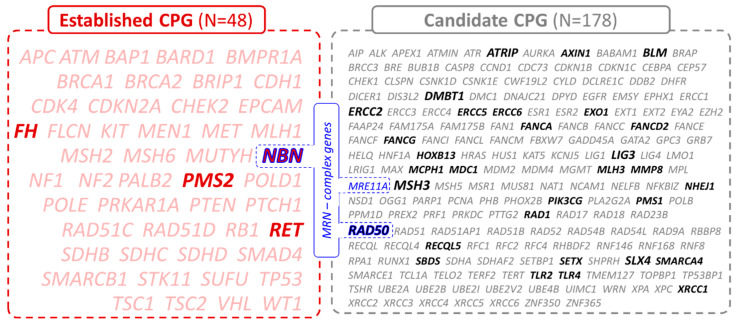
List of 226 cancer-predisposition genes divided into established (N = 48, in red) and candidate (N = 178, in grey) CPGs based on their clinical significance. The genes of the MRN complex are highlighted in blue. The PVs in CPGs highlighted in bold were found in this study.

**Figure 2 cancers-15-00201-f002:**
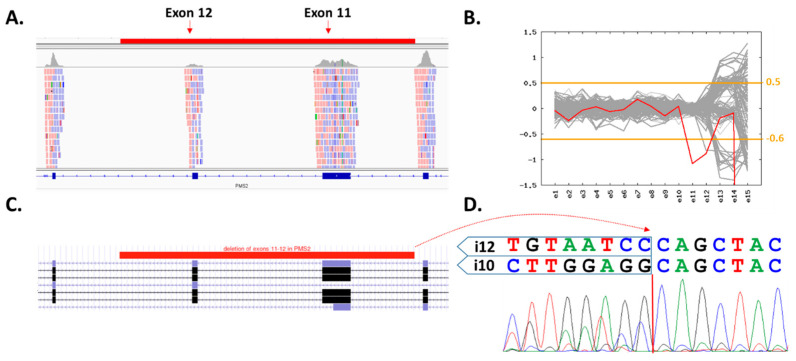
Characterization of exons 11–12 (8907 bp) PMS2 deletion. (**A**) Integrated Genome Viewer (IGV) visualization of the coverage decrease suggesting a presence of heterozygous two-exon deletion (highlighted region in red). (**B**) Visualization of CNV analyses generated using the CNVkit (patient’s sample in red). The inconsistent region covering exons 12–15 contains numerous PMS2 pseudogenes. (**C**) Schematic region of deletion in UCSC genome browser. (**D**) Characterization of the deletion breakpoint with Sanger sequencing.

**Table 1 cancers-15-00201-t001:** Clinicopathological characteristics in all 334 HCC patients.

Patients’ Characteristics	All Patients N = 334	Males N = 258	Females N = 76
Age [years]; median (range)	63 (26–77)	63 (36–75)	65 (26–77)
Cirrhosis; N (%)	329 (98.5)	254 (98.4)	75 (98.7)
Alcoholic	129 (38.6)	115 (44.6)	14 (18.4)
Viral	120 (35.9)	77 (29.8)	43 (56.6)
Cholestatic and autoimmune	48 (14.4)	37 (14.3)	11 (14.5)
NASH (non-alcoholic steatohepatitis)	29 (8.7)	23 (8.9)	6 (7.9)
Metabolic	3 (0.9)	2 (0.8)	1 (1.3)
Not present	5 (1.5)	4 (1.6)	1 (1.3)
HCC treatment; N (%)			
Liver transplantation	299 (89.5)	225 (87.2)	74 (97.4)
Other	35 (10.5)	33 (12.8)	2 (2.6)
HCC characteristics			
AFP [ng/mL]; median (range)	8.3 (0.9–5784)	7.4 (0.9–5784)	15.0 (1.9–1210)
Milan criteria; N (% of known)	220 (65.9)	168 (65.1)	52 (68.4)
Microangioinvasion; N (% of known)	128 (45.6)	91 (43.8)	37 (50.7)
Cholangiocarc. differentiation; N (% of known)	18 (5.4)	10 (4.6)	8 (10.8)
Grade 1; N (% of known)	37 (13.0)	22 (10.3)	15 (21.1)
Grade 2; N (% of known)	154 (54.0)	123 (57.5)	31 (43.7)
Grade 3; N (% of known)	94 (33.0)	69 (32.2)	25 (35.2)
Multiple primary tumor; N (%)	57 (17.1)	48 (18.7)	9 (11.9)
Malignancy in 1^st^/2^nd^ degree relatives; N (%)	131 (39.2)	99 (38.4)	32 (42.1)
Diabetes; N (%)	138 (41.3)	117 (45.3)	21 (27.6)
Obesity (BMI>30); N (%)	94 (28.1)	80 (31.0)	14 (18.4)
Smoking; N (%)	192 (57.5)	161 (62.4)	31 (40.8)

**Table 2 cancers-15-00201-t002:** Characterization of PV carriers with HCC in established high-to-moderate (**A**) and candidate (**B**) cancer predisposition genes. The PVs were present in the heterozygous state in all carriers.

Age at HCC Diagnosis (Years), Sex.	Variant	Personal ca History ^1^	Family ca History	Cirrhosis and HCC Features
**A. Established high-to-moderate cancer-predisposition genes**				
71, F	*PMS2*: c.1144+250_2175-1948del8907	0	0	Viral
72, M	*NBN*: c.657del5 (p.Lys219fs)	PrC (post)	PrC (father)	Alcoholic
68, M	*NBN*: c.657del5 (p.Lys219fs)	0	TC (father)	Alcoholic
68, M	*NBN*: c.657del5 (p.Lys219fs)	PrC (post)	0	Autoimmune
58, F	*NBN*: c.657del5 (p.Lys219fs)	0	0	Viral, HCC recurrence post-Tx
53, F	*FH*: c.1127A>C (p.Gln376Pro)	0	0	Viral
71, M	*RET*: c.2304G>C (p.Glu768Asp)	CRC (pre)	BC (mother); sarcoma (brother)	Alcoholic
**B. Candidate cancer-predisposition genes**				
69, F	*DMBT1*: c.2177-2A>C	BC	0	Autoimmune
66, M	*DMBT1*: c.4828+1G>A	0	Leu (brother); HNC (brother)	Alcoholic
69, M	*DMBT1*: c.4611C>G (p.Tyr1537Ter)	0	H&N (brother)	Alcoholic
71, M	*RAD50*: c.1875C>G (p.Tyr625Ter)	PrC (post)	BC (mother)	Viral
65, M	*RAD50*: c.2043delC (p.Val683fs)	0	0	Autoimmune
65, M	*RAD50*: c.2521del9 (p.Thr841fs) *	0	0	Viral, HCC recurrence post-Tx
65, M	*ATRIP*: c.1870del2 (p.Cys624fs)	0	0	Viral, HCC recurrence post-Tx
61, F	*ATRIP*: c.1152del4 (p.Gly385fs) *	0	0	Autoimmune
65, M	*BLM*: c.1642C>T (p.Gln548Ter)	0	LC (father)	Non-cirrhotic
64, M	*BLM*: c.1642C>T (p.Gln548Ter)	0	0	Autoimmune
68, M	*ERCC2*: c.2150C>G (r.2144_2190del45)	0	0	Autoimmune
53, M	*ERCC2*: c.2150C>G (r.2144_2190del45)	0	0	Viral, HCC recurrence post-Tx
71, F	*LIG3*: c.1283delT (p.His428fs)	0	0	Viral, HCC recurrence post-Tx
77, F	*LIG3*: c.799C>T (p.Arg267Ter)	0	LC (father)	Non-cirrhotic
68, M	*MSH3*: c.2686G>T (p.Gly896Ter)	0	Mel (mother)	Alcoholic
66, M	*MSH3*: c.1480delA (p.Asn494fs)	0	0	Alcoholic
66, F	*SLX4*: c.4207G>T (p.Glu1403Ter)	SkC (post)	0	Viral
58, M	*SLX4*: c.4024delA (p.Ser1342fs)	0	GaC (mother)	Autoimmune
57, F	*AXIN1*: c.64C>T (p.Arg22Ter)	0	0	Viral
62, M	*ERCC5*: c.3285del10 (p.Ser1096fs)	0	0	Viral, HCC recurrence post-Tx
71, M	*ERCC6*: c.537T>A (p.Tyr179Ter)	0	0	Viral
61, M	*EXO1*: c.1578del2 (p.Asp526fs)	CRC (pre)	0	NASH
60, F	*FANCA*: del16-17	0	0	NASH
59, M	*FANCD2*: c.990-1G>A	Leu (post)	GaC (mother)	NASH
60, F	*FANCG*: c.313G>T (p.Glu105Ter)	0	H&N (father); HCC (mother)	NASH
64, M	*HOXB13*: c.251G>A (p.Gly84Glu)	0	0	Autoimmune
59, M	*MCPH1*: c.126del2 (p.Phe43fs)	0	BC (mother)	Autoimmune, HCC recurrence post-Tx
57, M	*MDC1*: c.6081delC (p.Ser2028fs)	0	0	Alcoholic
40, M	*MMP8*: c.460G>T (p.Gly154Ter)	0	0	Viral
61, M	*MLH3*: c.3393dup2 (p.Thr1132fs)	0	BC (mother)	Autoimmune, HCC recurrence post-Tx
59, F	*NHEJ1*: c.169C>T (p.Arg57Ter)	0	HCC (mother)	Alcoholic
65, M	*PIK3CG*: c.2519del2 (p.Gln840fs)	H&N; SkC; PrC (all pre)	LC (father)	Alcoholic
54, M	*PMS1*: c.1009insA (p.Tyr337fs)	0	0	Viral
60, M	*RAD1*: c.168del5 (p.Lys57fs)	0	0	Alcoholic
41, F	*RECQL5*: c.2308C>T (p.Arg770Ter)	0	RCC (mother)	Viral
73, F	*SBDS*: c.258+2T>C	0	0	Viral
69, M	*SETX*: c.5074dup2 (p.Leu1692fs)	0	LC (father)	NASH
59, M	*SMARCA4*: c.859+1G>A	0	0	Viral
69, M	*TLR2*: c.1339C>T (p.Arg447Ter)	Lym (pre)	HCC (father)	Viral
74, M	*TLR4*: c.261-1G>C	0	BC (mother)	NASH
45, F	*XRCC1*: c.406dupT (p.Tyr136fs)	0	BC (mother)	Alcoholic

^1^ pre-/post- HCC diagnosis; * double variant carrier. Abbreviations: BC: breast cancer; ca: cancer; CRC: colorectal cancer; F: female; GaC: gastric cancer; H&N: head and neck cancer; HCC: hepatocellular carcinoma; Leu: leukemia; Lym: lymphoma; LC: lung cancer; M: male; Mel: melanoma; NASH: non-alcoholic steatohepatitis; PrC: prostate cancer; RCC: renal cell carcinoma; SkC: skin cancer except melanoma; TC: testicular cancer; Tx: transplantation.

**Table 3 cancers-15-00201-t003:** Germline PV identified in **A**. established high-to-moderate and **B**. candidate cancer predisposition genes. Statistically significant differences between variant frequencies of patients and controls are highlighted in bold.

Gene	Carriers in 334 Patients; N (%)	Carriers in 1662 PMC; N (%)	Odds Ratio (95% Confidence Interval)	*p*-Value
**A. Established high-to-moderate cancer-predisposition genes**				
*PMS2*	1 (0.3)	4 (0.3)	1.2 (0.14–11.11)	0.8
* **NBN** *	**4 (1.2)**	**4 (0.3)**	**5.0 (1.25–20.17)**	**0.012**
*FH*	1 (0.3)	0	n.d.	n.d.
*RET*	1 (0.3)	2 (0.1)	2.5 (0.23–27.49)	0.4
All carriers.	7 (2.1)	10 (0.6) ^1^		
**B. Candidate cancer-predisposition genes**				
* **DMBT1** *	**3 (0.9)**	**2 (0.1)**	**7.5 (1.25–45.13)**	**0.010**
* **RAD50** * ^2^	**3 (0.9)**	**3 (0.2)**	**5.0 (1.01–24.90)**	**0.029**
*ATRIP* ^2^	2 (0.6)	3 (0.2)	3.3 (0.56–19.98)	0.2
*BLM*	2 (0.6)	7 (0.4)	1.4 (0.30–6.87)	0.7
*ERCC2*	2 (0.6)	8 (0.5)	1.2 (0.26–5.88)	0.8
* **LIG3** *	**2 (0.6)**	**1 (0.1)**	**10.0 (0.91–110.48)**	**0.021**
*MSH3*	2 (0.6)	6 (0.4)	1.7 (0.33–8.26)	0.5
*SLX4*	2 (0.6)	2 (0.1)	5.0 (0.70–35.56)	0.1
*AXIN1*	1 (0.3)	0	n.d.	n.d.
*ERCC5*	1 (0.3)	0	n.d.	n.d.
*ERCC6*	1 (0.3)	0	n.d.	n.d.
*EXO1*	1 (0.3)	2 (0.1)	2.5 (0.23–27.49)	0.4
*FANCA*	1 (0.3)	7 (0.4)	0.7 (0.09–5.77)	0.7
*FANCD2*	1 (0.3)	0	n.d.	n.d.
*FANCG*	1 (0.3)	2 (0.1)	2.5 (0.23–27.49)	0.4
*HOXB13*	1 (0.3)	4 (0.3)	1.2 (0.14–11.14)	0.8
*MCPH1*	1 (0.3)	10 (0.6)	0.5 (0.06–3.88)	0.5
*MDC1*	1 (0.3)	0	n.d.	n.d.
*MLH3*	1 (0.3)	1 (0.1)	5.0 (0.31–79.75)	0.2
*MMP8*	1 (0.3)	5 (0.3)	1.0 (0.12–8.52)	0.9
*NHEJ1*	1 (0.3)	0	n.d.	n.d.
*PIK3CG*	1 (0.3)	0	n.d.	n.d.
*PMS1*	1 (0.3)	2 (0.1)	2.5 (0.23–27.49)	0.4
*RAD1*	1 (0.3)	0	n.d.	n.d.
*RECQL5*	1 (0.3)	6 (0.4)	0.8 (0.10–6.89)	0.9
*SBDS*	1 (0.3)	13 (0.8)	0.4 (0.05–2.91)	0.3
*SETX*	1 (0.3)	10 (0.6)	0.5 (0.06–3.88)	0.5
*SMARCA4*	1 (0.3)	0	n.d.	n.d.
*TLR2*	1 (0.3)	1 (0.1)	5.0 (0.31–79.75)	0.2
*TLR4*	1 (0.3)	2 (0.1)	2.5 (0.23–27.49)	0.4
*XRCC1*	1 (0.3)	7 (0.4)	0.7 (0.09–5.77)	0.7
All carriers	40 (12.0)	104 (6.3) ^1^	-	-

^1^ only frequency of germline variants in genes found in HCC group is shown. ^2^ a patient carrying simultaneous RAD50 and ATRIP germline variant. n.d.: not defined. Statistically significant differences between variant frequencies of patients and controls are highlighted in bold.

**Table 4 cancers-15-00201-t004:** Comparison of germline panel studies in HCC patients. The table describes only genes that were analyzed in at least two cohorts and where a carrier of heterozygous PV was found. Established high-to-moderate cancer predisposition genes are highlighted in bold letters.

	This Study; N (%)	Uson Junior et al. 2022 (Ref. [9]); N (%)	Mezina et al. 2021 (Prospective) (Ref. [10]); N (%)	Mezina et al. 2021 (Retrospective) (Ref. [10]); N (%)
HCC patients analyzed (N)	334	44	217	219
Genes analyzed (N)	226	83	134	1–154
***APC***	**0**	**0**	**0**	**2 (0.91)**
***ATM***	**0**	**0**	**0**	**1 (0.46)**
***BARD1***	**0**	**1 (2.27)**	**0**	**1 (0.46)**
*BLM*	2 (0.59)	0	0	0
***BRCA1***	**0**	**0**	**0**	**1 (0.46)**
***BRCA2***	**0**	**0**	**2 (0.92)**	**6 (2.74)**
***BRIP1***	**0**	**0**	**4 * (1.84)**	**1 (0.46)**
***CDKN2A***	**0**	**1 (2.27)**	**0**	**0**
***CHEK2***	**0**	**0**	**3 * (1.38)**	**2 (0.91)**
*FANCA*	1 (0.29)	n.a.	5 (2.30)	1 (0.46)
*FANCD2*	1 (0.29)	n.a.	2 (0.92)	0
*FANCG*	1 (0.29)	n.a.	0	0
*FANCM*	0	n.a.	1 * (0.46)	0
***FH***	**1 (0.29)**	**0**	**0**	**2 (0.91)**
*HOXB13*	1 (0.29)	0	0	0
*MITF*	n.a.	1 (2.27)	1 * (0.46)	1 (0.46)
*MLH3*	1 (0.29)	n.a.	0	0
***MSH2***	**0**	**0**	**0**	**2 (0.91)**
*MSH3*	2 (0.59)	n.a.	1 * (0.46)	0
***MSH6***	**0**	**0**	**1 (0.46)**	**0**
*MUTYH*	0	0	3 (1.38)	2 (0.91)
***NBN***	**4 (1.19)**	**2 (4.54)**	**0**	**2 (0.91)**
***NF1***	**0**	**0**	**1 (0.46)**	**0**
*NTHL1*	n.a.	0	0	1 (0.46)
***PALB2***	**0**	**0**	**0**	**3 (1.37)**
***PMS2***	**1 (0.29)**	**0**	**1 (0.46)**	**0**
*RAD50*	3 * (0.89)	1 (2.27)	1 (0.46)	0
***RAD51D***	**0**	**1 (2.27)**	**0**	**0**
***RET***	**1 (0.29)**	**0**	**0**	**0**
*SLX4*	2 (0.59)	n.a.	0	0
*SMARCA4*	1 (0.29)	0	0	0
*TMEM127*	0	0	1 * (0.46)	0
***TP53***	**0**	**0**	**0**	**2 (0.91)**
* **Established PV carriers in CPG** * *****	**7 (2.1)**	**5 (11.4)**	**12 (5.5)**	**25 (11.4)**
* **All carriers (referred in the study; N)** *	47 (14.1)	7 (15.9)	25 (11.5)	30 (13.7)

n.a.—not analyzed; * double-variant carrier of RAD50 and ATRIP (not shown in this table) in this study (Table 2); CHEK2 and BRIP1, FANCM and TMEM127, MITF and MSH3 in [8]. Established high-to-moderate cancer predisposition genes are highlighted in bold letters.

## Data Availability

The data generated and analyzed in this study are included in this publication. Identified variants were submitted to ClinVars under accession no. SCV002569169–SCV002569175, or are available from the corresponding author upon reasonable request.

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
