# Peer review of "Low Frequency of Cancer-Predisposition Gene Mutations in Liver Transplant Candidates with Hepatocellular Carcinoma"

_cancers, 2022, doi:10.3390/cancers15010201_

Round 1

Reviewer 1 Report

Thank you for the opportunity to review the manuscript “Low frequency of cancer-predisposition gene mutations in liver transplant candidates with hepatocellular carcinoma” by Horackova et al. The authors examined a cohort of patients with early-stage HCC who were candidates for liver transplantation for cancer-associated germline mutations.  The authors confirm the rarity of PVs in early-stage HCC while partially reporting on patient outcomes post-transplantation.  Structing these observations into an additional table and potentially extending these observations to de novo malignancies would substantially increase the impact of the manuscript.  Specific comments are bulleted below:

·       - The manuscript will require editing for English language to improve clarity.

·         - The authors identified several high-to-moderate and candidate pathogenic variants in the HCC population, but there does not appear to be a relationship between any individual PV and HCC recurrence post-transplant. The unstructured observations regarding post-transplant HCC recurrence in Table 2 raise a couple additional questions:

o   the authors should provide a breakdown for how many patients were treated with liver transplantation. 

o   It would also be interesting to see the breakdown of PVs in transplant patients, including both HCC recurrence as well as de novo malignancy post-transplant.

·         - Since these are germline variants, it would be interesting to include in Table 4 a race/ethnicity/geographic breakdown of the individual cohorts.

Reviewer 2 Report

The manuscript nicely described the prevalence of pathogenic variants in cancer-predisposition genes in HCC patients considered for liver transplantation. Overall, the discussion included good comparison to other papers that did similar studies, However, more discussion on how absence of these mutations in a subset of HCC patients can truly justify not to screen the patients for germline genetic screening might be necessary.
